# Privacy Auditing of Large Language Models

**Ashwinee Panda**[p*] **Xinyu Tang**[p*] **Milad Nasr**[g] **Christopher A. Choquette-Choo**[g] **Prateek Mittal**[p]
[p]Princeton University, [g]Google DeepMind, [*]Equal contribution

## Abstract

Current techniques for privacy auditing of large language models (LLMs) have limited efficacy—they rely on basic approaches to generate canaries which leads to weak membership inference attacks that in turn give loose lower bounds on the empirical privacy leakage. We develop canaries that are far more effective than those used in prior work under threat models that cover a range of realistic settings. We demonstrate through extensive experiments on multiple families of fine-tuned LLMs that our approach sets a new standard for detection of privacy leakage. For measuring the memorization rate of non-privately trained LLMs, our designed canaries surpass prior approaches. For example, on the Qwen2.5-0.5B model, our designed canaries achieve $49.6\%$ TPR at $1\%$ FPR, vastly surpassing the prior approach's $4.2\%$ TPR at $1\%$ FPR. Our method can be used to provide a privacy audit of $\varepsilon \approx 1$ for a model trained with theoretical $\varepsilon$ of 4. To the best of our knowledge, this is the first time that a privacy audit of LLM training has achieved nontrivial auditing success in the setting where the attacker cannot train shadow models, insert gradient canaries, or access the model at every iteration.

## 1 Introduction

Despite the growing success of massively pretrained Large Language Models (Brown et al., 2020; OpenAI, 2023; Gemini-Team et al., 2023), there is also growing concern around the privacy risks of their deployment (McCallum, 2023; Bloomberg, 2023; Politico, 2023), because they can memorize some of their training data verbatim (Carlini et al., 2019; 2021; 2023b; Biderman et al., 2023a).

There is currently a discrepancy between memorization studies in large frontier models reports that show very limited memorization and a line of research showing that data can be extracted from such models (Carlini et al., 2021; 2023a; Nasr et al., 2025). With the goal of understanding concerns around the privacy risks of deploying LLMs, currently, model developers study the quantifiable memorization of their models by inserting canary sequences and testing for memorization, and they conclude that the models do not memorize much (Anil et al., 2023; Reid et al., 2024).

The gap between these two bodies of work is in the data being memorized. When developers insert canaries, they are not necessarily inserting the canaries that are most likely to be memorized. However, when researchers try to extract data, they are extracting the "most extractable" data, which by definition was the most likely to be memorized. Without better design of canaries, model developers will systematically underestimate the privacy leakage of their models. In this work, we aim to develop stronger privacy audits by developing canaries that are more likely to be memorized.

We are primarily interested in understanding privacy leakage from LLMs through the lens of membership leakage of a canary dataset used in training an LLM (used to measure the privacy leakage). Specifically, we want to understand how to construct the most easily memorized canaries for language models. Qualitatively, if we find that membership information attacks (MIA) on these canaries for LLMs can be very effective, this improves our understanding of the privacy leakage of LLMs.

Membership inference attacks are also used in auditing the privacy of differentially private models. The effectiveness of privacy auditing hinges on the selection of optimal "canaries". We introduce new methods for generating *easy-to-memorize* input space canaries, and use these to improve the performance of existing privacy auditing methods and obtain tighter empirical bounds on privacy leakage. We provide the first privacy audit for the black-box setting for LLMs. Our audit achieves a non-trivial lower bound of $\varepsilon \approx 1$ for a model trained to an upper bound of $\varepsilon = 4$.

## 2 BACKGROUND

### 2.1 MEMBERSHIP INFERENCE ATTACKS

Membership inference attacks (MIAs) (Shokri et al., 2017) are one of the simplest privacy threats in machine learning: the goal is to predict whether a specific example was part of a model's training set (member) or not (non-member). MIAs exploit differences in model behavior on members vs non-members, using signals such as the target sample's loss (Yeom et al., 2018), the loss of neighboring samples (Mattern et al., 2023), or information from reference models (Carlini et al., 2021).

The primary goal of our work is to estimate privacy leakage in models, independent of developing new MIAs. Evaluating MIAs on synthetic canaries inserted into LLM training can inform both memorization and generalization in LLMs (Gemini-Team et al., 2023; Reid et al., 2024; Anil et al., 2023). With $\mathbb{1}$ as the indicator function, $\tau$ a tunable threshold, and $\mathcal{A}'$ a confidence score function (in Yeom et al. (2018) this is the model loss), membership is predicted as: $\mathcal{A}(x, y) = \mathbb{1}[\mathcal{A}'(x, y) > \tau]$.

Recently, Duan et al. (2024) evaluated a range of MIAs (Yeom et al., 2018; Carlini et al., 2021; Mattern et al., 2023; Shi et al., 2024) against large language models (LLMs) and found that MIAs are largely ineffective in this context. They attribute this to factors such as the single-epoch training typically used in LLMs. They argue that realistic MIA evaluations require high overlap between members and non-members. However, prior work has often achieved MIA success by exploiting distribution shifts between these groups. Related studies (Meeus et al., 2024; Das et al., 2024; Eichler et al., 2024) confirm that distribution shift is the primary driver of MIA success.

In our work, our sampling process for member and non-member datapoints is IID across the dataset that we draw them from. We detail this dataset in each section: in Section 4, this is validation data and in Section 5, this dataset is random tokens. Therefore, the problem of distribution shifts identified in Meeus et al. (2024); Duan et al. (2024) does not exist. This is different from prior work, which requires the IID property to hold across the entire pretraining dataset that they consider.

There are three main avenues for improving privacy audits: (1) selecting more separable data, (2) using better statistics, and (3) designing improved tests based on those statistics. While prior work extensively explored (2) and (3) without much success, Duan et al. (2024) showed that current MIAs cannot reliably distinguish member from non-member data in LLMs. Our work focuses on (1), demonstrating that selecting more separable data alone enables strong privacy audits, even when using the simple loss-based attack proposed by Yeom et al. (2018). Our contribution is complementary to future work on developing new MIAs, which could leverage our techniques.

### 2.2 AUDITING DIFFERENTIALLY PRIVATE LANGUAGE MODELS

We provide a concise overview of differential privacy (DP), private machine learning, and methods to audit the privacy assurances claimed under DP. Differential privacy is the gold standard for providing a provable upper bound on the privacy leakage of an algorithm (Dwork et al., 2006).

**Definition 2.1** (($\varepsilon, \delta$)− Differential Privacy (DP)). Let $\mathcal{D} \in \mathcal{D}^n$ be an input dataset to an algorithm, and $\mathcal{D}'$ be a neighboring dataset that differs from $D$ by one element. An algorithm $\mathcal{M}$ that operates on $\mathcal{D}$ and outputs a result in $S \subseteq \text{Range}(\mathcal{M})$ is considered to be ($\varepsilon, \delta$)-DP if: For all sets of events $S$ and all neighboring datasets $D, D'$, the following holds:

$$\Pr[\mathcal{M}(D) \in S] \le e^{\varepsilon} \Pr[\mathcal{M}(D') \in S] + \delta \tag{1}$$

**Differentially Private Machine Learning.** Differentially Private Stochastic Gradient Descent (DP-SGD) (Song et al., 2013; Abadi et al., 2016) is the workhorse method for training neural networks on private data.

**Definition 2.2** (Differentially Private Stochastic Gradient Descent (DP-SGD)). For a batch size $B$, learning rate $\eta$, clipping threshold $C$, and added noise standard deviation $\sigma$, the DP-SGD update rule at iteration $t$ on weights $w$ is given by:

$$w^{(t+1)} = w^{(t)} - \frac{\eta}{|B|} \left( \sum_{i \in B} \frac{1}{C} \mathbf{clip}_{\mathrm{C}}(\nabla \ell(x_i, w^{(t)})) + \sigma \xi \right) \tag{2}$$

DP-SGD does per-sample gradient clipping on top of SGD to limit the sensitivity of each sample, and adds noise sampled i.i.d. from a $d$-dimensional normal distribution with standard deviation $\sigma$, $\xi \sim \mathcal{N}(0, I_d)$.

**Auditing DP-SGD.** DP guarantees are expressed in terms of a failure probability $\delta$ and a privacy budget $\varepsilon$. In machine learning, we can interpret the DP guarantee as an upper bound in terms of $e^\varepsilon$ on the adversary's success rate in membership inference that holds with probability $1 - \delta$. As shown by Kairouz et al. (2015), if $\mathcal{M}$ is $(\varepsilon, \delta)$-DP, it defines a *privacy region* such that an attacker's TPR and FPR (also Type I $\alpha$ and Type II $\beta$ errors) cannot exceed the bounds of this region, given by

**Definition 2.3** (Privacy Region of $(\varepsilon, \delta)$-DP (Kairouz et al., 2015))**.** if $\mathcal{M}$ satisfies $(\varepsilon, \delta)$-DP, then it establishes a privacy region that bounds any adversary's type I ($\alpha$) and type II ($\beta$) errors. The privacy region is define as follow:

$$\begin{aligned}
\mathcal{R}(\varepsilon, \delta) = \{(\alpha, \beta) \mid &\alpha + e^\varepsilon \beta \geq 1 - \delta \wedge e^\varepsilon \alpha + \beta \geq 1 - \delta \wedge \\
&\alpha + e^\varepsilon \beta \leq e^\varepsilon + \delta \wedge e^\varepsilon \alpha + \beta \leq e^\varepsilon + \delta\}
\end{aligned} \tag{3}$$

For differentially private machine learning, our objective in privacy auditing is to provide an empirical lower bound on the privacy leakage from an algorithm $\mathcal{M}$. Privacy audits are useful because they give us information about how tight the upper bound is that we obtain from DP (Steinke et al., 2023), and if the privacy audit produces a lower bound that is greater than the upper bound given by DP-SGD, we can use this to find errors in the DP-SGD implementation (Tramer et al., 2022).

Steinke et al. (2023) propose a recent privacy auditing method that we use in this paper, which can provide an audit without needing to train multiple models. However, they are not able to provide a nontrivial result when training on real data in the black-box setting (where the canaries exist in the input space and the attacker observes the loss of the model), and do not provide audits for language models (they only provide audits for computer vision).

**Summary of DP Background.** DP-SGD provides a mathematical proof that gives an upper bound on the privacy parameter. A privacy audit is a procedure that provides a lower bound on the privacy parameter. Privacy audits can be used to ascertain the correctness of DP-SGD training and estimate the tightness of analysis. Many privacy auditing methods have been proposed, but no privacy auditing method has been able to provide a nontrivial lower bound of an LLM trained with a realistic DP guarantee ($\varepsilon < 10$ on real data in the black-box setting in a single run).

## 3 CRAFTING CANARIES THAT ARE EASY TO SPOT

Previous research has consistently shown that some out-of-distribution (OOD) inputs are more prone to memorization by machine learning models (Carlini et al., 2022a; Nasr et al., 2021; 2023; Carlini et al., 2022b). Leveraging this insight, existing methods for generating canaries in membership inference attacks often focus on crafting OOD inputs so that they have a higher likelihood of being memorized. In the context of large language models (LLMs), creating out-of-distribution (OOD) inputs typically involves using random tokens. These inputs are assumed to be anomalies that the model will easily memorized. However, previous works (Carlini et al., 2022a; Nasr et al., 2023) have shown that not all OOD examples are easily learned and memorized by the model. There is a wide range of OOD examples that can be used in membership inference attacks. While basic approaches have shown some success, there is potential for significant improvement.

To improve over this *random canary baseline*, we will show how an adversary can attack the tokenizer to create canaries that are easier to spot (see Section 3.2). Next, we define what we mean by a canary.

### 3.1 THE CANARY SETUP

A canary is the concatenation of two sequences of tokens: a prefix and a secret both sampled from some randomness (Carlini et al., 2019).

**MIA method.** All current MIAs for LLMs require the loss (Duan et al., 2024); thus, as we discussed in Section 2, we use the simplest loss thresholding attack of Yeom et al. (2018) which predicts all points (canaries) with loss less than or equal to some learned value $\tau$ as a member, and the rest as non-members. Because our approach works with the simplest MIA, we expect it will generalize. The

*loss calculation* depends on the training objective for the target model. We calculate the loss on all trainable tokens of the sequence, i.e., just for the canary tokens in prefix-learning and for the entire sequence (including the prefix) in next word prediction (objectives detailed more below).

**Training objective.** We consider standard objectives for both supervised fine-tuning and pretraining. For fine-tuning, we consider prefix language modeling (Raffel et al., 2020) which masks out the loss on the prefix that we do not want the model to learn. Figure 1 shows the results for this objective. For pretraining, we consider a next word prediction (NWP) objective where the model is trained to predict each next token in the sequence in parallel via teacher-forcing. Figure 2 shows these results.

**Comparing attack efficacy.** There are many ways to compare attack efficacy each with pros and cons. Following Carlini et al. (2022a), we use the true-positive rate (TPR) at low false-positive rate (FPR), for which we pick FPR=1%. When we audit DP, we use $\varepsilon$ lower bounds as is standard (Jagielski et al., 2020; Nasr et al., 2021; 2023; Steinke et al., 2023); these essentially define a region where the TPR and FPR must be bounded by Equation (3).

**Canary size.** Prior works (Anil et al., 2023; Gemini-Team et al., 2023) use many thousands of canaries, with prefixes and secrets each constructed from 50 random tokens. We find that we only need 1000 canaries for $3.6 \times 10^7$ tokens in our finetuning dataset. Because each canary is generally just a single token (secret) appended to a normal sample (prefix), just a small fraction (0.0027%) of our dataset is constituted of canaries.

**Selecting the canary prefix.** As we previously mentioned, we want to ensure that we sample canaries IID from some distribution so that our MIA success cannot be attributed simply to distribution shift, as in Duan et al. (2024). Each canary prefix is generated using one of 1000 unique samples from the test set; we use the test dataset for this to be more aligned with practical use cases where the prefix contains semantic information. For simplicity and because this is the most challenging setting, we use secrets that are one token in length. In Table 2, we show that our attacks still in general outperform the baseline even when the number of secret tokens is increased.

## 3.2 SOME CANARIES SING LOUDER THAN OTHERS

The most important part of our canary design is the algorithm by which we generate the secret. Our main intuition, as discussed at the beginning of Section 3, is to craft canaries that are easy to spot. An easy way to do this is with gradient-space canaries, but we don't have the freedom to do this because we only want to design the more difficult input-space canaries. Our strategy is to give the adversary increasing strength in terms of a priori knowledge of the training data distribution.

We begin by formalizing our goal. We desire a secret $x_t$ such that when given the prefix $x_{1:t-1}$ the model's loss $p(x_t|x_{1:t-1})$ is high, i.e., it is unlikely to have been seen under the model. Importantly, we must have an estimate on this priori, i.e., before training the model $p$, as we will be injecting these canaries into model training for auditing.

With this in mind, it is clear why **random canaries** (Anil et al., 2023; Gemini-Team et al., 2023), i.e,. canaries with randomly chosen secrets are a strong baseline. A weak adversary with no knowledge of the data distribution a priori can at best choose a random secret as this maximizes its entropy in the limit of long secrets. It is this baseline from prior work which we seek to beat, and which we will do so, by considering adversaries with increasing knowledge of the training data distribution a priori.

**How to make adversaries stronger.** First, recall that our goal is to design strong *privacy audits*. A privacy audit, as discussed in Section 2.2, is a tool that model developers use to estimate the worst-case privacy leakage, as measured by a lower-bound on the observed privacy leakage $\epsilon$. When audits can be trusted to be close to a ground-truth upper-bound (i.e., when DP training is used), they can give a model developer faith that a model is private.

Privacy audits use the membership inference attack as a core component, and use the ROC curve to get a lower bound on epsilon. But, because this audit is run by a model developer, and not by a third-party attacker, adversaries should be assumed to be (reasonably) strong so as to adequately measure the worst-case. For this reason, and as motivated above, we make the adversary stronger by giving them a prior knowledge of the training data distribution. Notice that this is not unreasonable: LLMs are trained on the web and this data is publicly accessible. When models are fine-tuned on private data, there may still exist public surrogates that can strengthen an adversary in this way.

We next give three methods by which an adversary can estimate $p(x_t|x_{1:t-1})$ a priori.

**Unigram canaries.**[1] Given an approximate list of frequencies of tokens in the dataset, or in other words a unigram model, the attacker can select the least common tokens and use them as secrets in canaries. As we can see in Figure 1 ('unigram'), this works quite well.

**N-gram Canaries.** Naturally, if we want to insert longer canaries, we can use an N-gram model instead of a unigram to generate canaries. If we fit a bigram model, we can generate the pair of tokens $x, y$ such that $y$ is unlikely to follow $x$ and $x$ is unlikely to follow the preceding token in the document where it was inserted. We present the 'bigram' results in Figure 1.

**Model-Based Canaries.** A potential flaw in the above strategies is that they only account for the distribution of the training dataset and not of the model's distribution. If we want to audit finetuning, then we may need to consider not only what tokens are seldom seen in the finetuning dataset but also what tokens the model itself is unlikely to generate. If the attacker has black-box access to the model before they insert the canary, they can just query the model to get the least likely continuation of their prefix. However, this requires training two models or approximating it using a past model.

### 3.3  CANARIES VIA NEW TOKENS

Our underlying insight is that examples can be easily identified as members by the presence of tokens that do not appear anywhere else in the training dataset. The embedding table in a language model is both large, with, e.g., output dimension $151,936$ (Qwen-Team, 2024), and receives only a sparse update for only the tokens seen in training. Thus, a model that has not received a gradient for a given row will behave very differently when predicting that token than a model that has.

We consider the setting where a model developer wants to understand the worst case privacy leakage of their model training, as in Chowdhery et al. (2022); Anil et al. (2023); Reid et al. (2024). We take advantage of the model developer's direct access to the model to easily craft canaries that are guaranteed to have high loss (low $p(x_t|x_{1:t-1})$) for any prefix instead of relying on heuristics. The model developer can simply introduce *new tokens* that have never been seen by the model before, are only used in the canary secrets, and are therefore always going to have high loss. This is similar to other special tokens that are used in training, e.g., control tokens that are reserved for later use. Indeed, many recent LLMs are released with special tokens present in the embedding that are untrained, e.g., Mistral (Jiang et al., 2023) and LLama (Touvron et al., 2023). Note that once trained, the rows of the embedding matrix corresponding to these tokens can be easily removed or reinitialized without affecting the model utility significantly.

As we show in Figure 1, introducing new tokens is an incredibly effective way to generate canaries that can be used during pretraining without any accuracy degradation (the 'new' column). While new token canaries contain less semantic information than other canaries in measuring the memorization rate of LLMs because new tokens are added without concrete semantic information, this is a valid privacy audit because the DP-SGD guarantees hold not only for random initialization but also for *any fixed initialization*. We are generating these canaries to be as strong as possible, including in the setting of DP, which is the most useful thing because we can now audit DP-SGD.

## 4  A SYSTEMATIC EVALUATION OF MEMORIZATION IN LLM TRAINING

**Models.** We use our designed canaries to evaluate the memorization rate across a wide range of model series. We consider **4** model series and **10** models in total including GPT2 (Radford et al., 2019), Pythia (Biderman et al., 2023b)], Qwen-2.5 (Qwen-Team et al., 2024; Qwen-Team, 2024), and Llama3 (Team et al., 2024). More details are in Appendix A. Our chosen set of models also spans the range of vocabulary sizes from 50k (GPT2, Pythia), 128k (Llama), up to 150k (Qwen), validating that our methods are viable for all vocabulary sizes used in models today. Though prior works have considered GPT2 (Li et al., 2022; Yu et al., 2022), we are also interested in more powerful models like Llama and Qwen because they are used in practice and understanding how easily they memorize data can help us better understand how to audit frontier models.

---

[1] Herein, we use 'gram' to mean token, despite it historically meaning characters.

**Datasets.** We finetuned the models on PersonaChat (Zhang et al., 2018) and E2E (Novikova et al., 2017), which are used for DP evaluations in prior works (Li et al., 2022; Yu et al., 2022; Panda et al., 2024). PersonaChat is a dataset that consists of conversations of people describing themselves. E2E dataset is a natural language generation task that maps restaurant template information to reviews. All experiments were conducted on a single A100 GPU. We finetuned models on these two datasets with a canary sub-sampling rate $q = 0.01$ and steps $T = 100$ to approximate the setting of single-epoch training on the canary set. Note that this is a more challenging task as Duan et al. (2024) argue that single-epoch training is one reason why membership inference is difficult in LLMs.

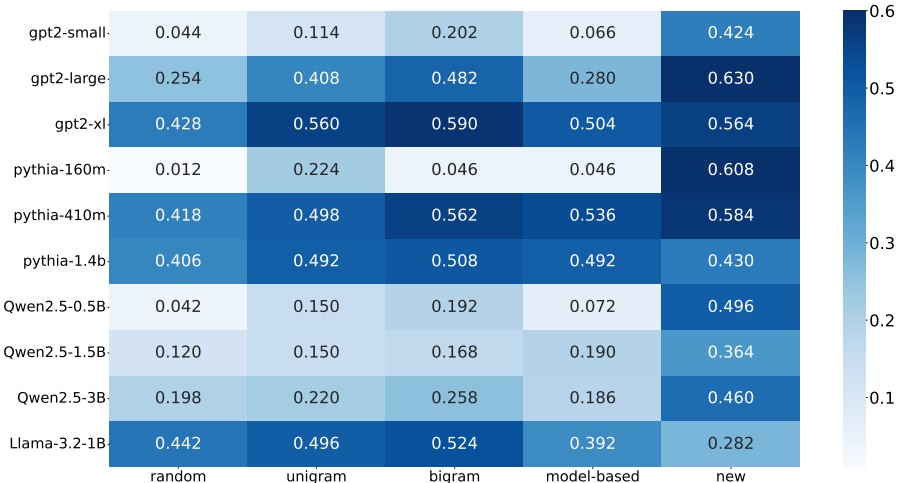

Figure 1: We visualize the True Positive Rate of the membership inference attack on PersonaChat at a low false positive rate of $1\%$. Our proposed canaries outperform the random canary.

**Results.** Figure 1 illustrates the True Positive Rate (TPR) of the membership inference attack (MIA) at $1\%$ False Positive Rate (FPR) for all canary crafting techniques across 3 model families and 3 sizes in each model family. Our proposed canaries consistently outperform the random canary baseline, with the new token canary performing consistently well across all model sizes. The unigram and binary canaries do better for larger models, which can accurately learn the N-gram priors we model with these approaches. We are particularly excited by the performance of the bigram canary approach, which performs well without needing to add new tokens into the vocabulary. Our results suggest that current reports of privacy leakage that only rely on the random canaries, e.g., those in Anil et al. (2023); Gemini-Team et al. (2023), may underestimate the privacy leakage.

We presented results in Figure 1 with a Supervised Finetuning (SFT) objective where the prefix is masked out and the gradient is only taken on the canary tokens. Finetuning tasks generally use an SFT loss. In Figure 2 we present results with a Next Word Prediction (NWP) objective, as would be used during pretraining. We find that this significantly decreases the effectiveness of the attack for the smaller models. However, for the larger models, the new token canary still works well.

In Table 1 we validate that our new token canary significantly outperforms the random canary baseline on the E2E dataset (Novikova et al., 2017) across the GPT and Pythia models. In Table 2 we increase the number of canary tokens that we append from 1 to 8 and find that this significantly increases the MIA success for both the unigram and random canaries. Intuitively, longer canaries are easier to tell apart. At 8 canary tokens, the unigram canary outperforms the random canary, indicating that our unigram approach has some merit. As we show in Appendix Figure 3, the unigram approach consistently selects sequences that are more OOD, as measured by frequency, than the random canary.

## 5 DP AUDITING EVALUATION

In Section 4, we showed the effectiveness of our attack for LLMs in the non-private setting, reporting the TPR at a low FPR. We now present privacy auditing results for models trained with DP-SGD,

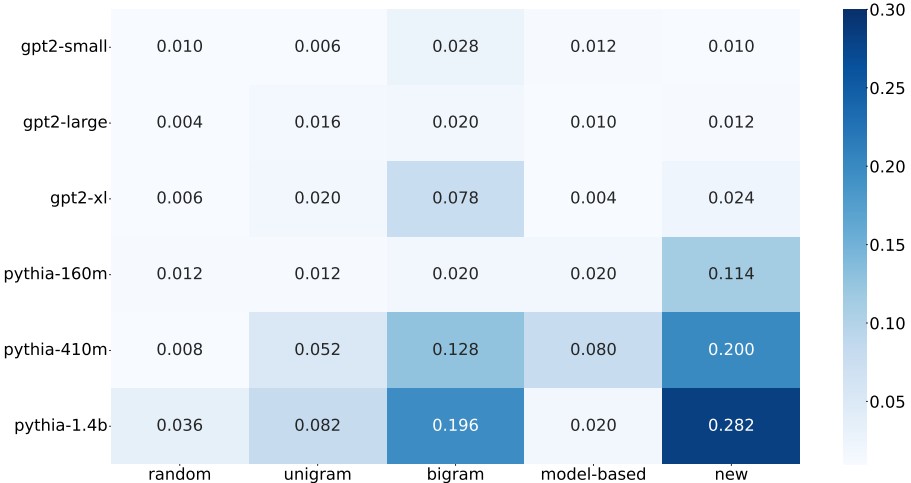

Figure 2: We replace the SFT loss used in Figure 1 with a NWP loss, on PersonaChat. MIA TPR is worse with a NWP loss, but our proposed bigram and new token canaries still outperform the random baseline.

Table 1: MIA results on E2E follow the trends on PersonaChat, with new beating random.

| Train Obj. | Canary | pythia | | | gpt2 | | |
|---|---|---|---|---|---|---|---|
| | | 160m | 410m | 1.4b | small | large | xl |
| NWP | new | **0.446** | **0.260** | **0.350** | **0.250** | **0.408** | **0.332** |
| | random | 0.012 | 0.014 | 0.072 | 0.006 | 0.004 | 0.010 |
| SFT | new | **0.586** | **0.654** | **0.643** | **0.572** | **0.622** | **0.654** |
| | random | 0.080 | 0.330 | 0.050 | 0.058 | 0.366 | 0.420 |

Table 2: Increasing the number of canary tokens significantly increases MIA success.

| # Tokens. | Canary | gpt2 | | | pythia | | |
|---|---|---|---|---|---|---|---|
| | | small | large | xl | 160m | 410m | 1.4b |
| 1 | unigram | 0.114 | 0.408 | 0.560 | 0.224 | 0.498 | 0.492 |
| | random | 0.044 | 0.254 | 0.428 | 0.012 | 0.418 | 0.406 |
| 8 | unigram | 0.386 | 0.568 | 0.590 | 0.264 | 0.592 | 0.614 |
| | random | 0.248 | 0.434 | 0.556 | 0.158 | 0.478 | 0.578 |

where we want to obtain the best lower bound on $\varepsilon$. We first discuss our auditing setup in Section 5.1. We then present our main auditing results in Section 5.2.

## 5.1 SETUP

We use the privacy auditing procedure of Steinke et al. (2023). This means that we randomly generate 1000 canaries, insert half of them, and try to do membership inference on the entire set. The accuracy of our MIA then translates into a lower bound with a 95% (or 99%) confidence interval on $\varepsilon$, meaning that the privacy loss is at least $\varepsilon$. This is the exact same implementation and confidence interval, etc. as in Steinke et al. (2023). One parameter in the method is the number of guesses that the adversary makes. We find that 100 guesses is sufficient to get a useful metric for DP auditing. For 100 guesses, the upper bound for empirical $\varepsilon$, i.e., getting 100 guesses correctly, is 2.99 for a 99% confidence interval and $\delta = 10^{-5}$. Our canaries are always randomly sampled IID from their distribution.

We use the following terminology from Nasr et al. (2023): the setting where the attacker has access to all intermediate steps is "white-box", and the setting where the attacker can only see the last iteration is "black-box." We always use the *black-box* setting where the attacker has to perform their audit only using the final trained model. Furthermore, we consider the setting where the attacker only has access to the logprobs of the final model given some input, and is not able to query the weights. This is the most realistic setting because it matches the access that ordinary users have to frontier models. Moreover, previous works (Morris et al., 2024; Carlini et al., 2024) show that it is possible for the attacker to evaluate the logprobs in settings where they are not directly outputted by the APIs.

In this black-box setting, the SOTA single-run privacy audit (Steinke et al., 2023) shows an empirical $\varepsilon \approx 1.3$ for analytical $\varepsilon = 4$ under a 95% confidence interval when auditing a ResNet trained on CIFAR10. We use this setting (1000 canaries, analytical $\varepsilon = 4$) for all of our privacy auditing experiments, but additionally report both the 95%, 99% confidence intervals. Our objective is to show that our method can recover a similar audit (in experimental results we achieve empirical $\varepsilon \approx 1.3$)

in the same setting, because there is no work that provides a method that can perform a nontrivial privacy audit of LLMs in this setting (Kazmi et al. (2024) do not provide a formal privacy audit).

**Changes from MIA.** In Section 4, we used prefixes randomly sampled from the validation set to construct our canaries. However, for DP auditing, we instead use prefixes composed of randomly sampled tokens to construct the canary. We find this design choice is essential to achieve non-trivial auditing results for DP training of LLMs in Table 8. We use an SFT loss function for DP auditing, because we found in the previous section that it leads to a much better MIA (Figure 1 vs. Figure 2), and indeed we validate that the SFT objective is critical for tight DP auditing in Table 9.

In this section, we train models with DP-SGD under $\varepsilon = 4$ for $T = 1000$ steps with a subsampling rate of $q = 0.1$. We report the empirical $\varepsilon$ estimation both in 95% (the main setting in Steinke et al. (2023)) and 99% confidence. By increasing the confidence level, we get a more conservative empirical $\varepsilon$ estimation. Across both confidence levels, our proposed token canaries gives a tighter empirical $\varepsilon$ estimation, i.e., more close to the theoretical $\varepsilon$ (higher is better), than the random canary baseline.

## 5.2 EVALUATION

**Main Results.** We present our main results for auditing DP-SGD in Table 3, where we train GPT2-small. We train on both PersonaChat and the E2E dataset, with FFT and LoRA. We find that LoRA finetuning obtains similar auditing performance to FFT, with worse perplexity. We tried ranks between 4 and 256 and found little difference, so we report results with rank 8. Auditing results are also similar across datasets; at a 99% CI, the new token canary gives us an audited $\varepsilon$ of 1.01 for both FFT on PersonaChat and LoRA on E2E. This indicates that our new token canary can be used for an effective audit on different datasets. Compared to the random canary baseline, our proposed canary strategies achieve far better privacy estimation

Table 3: We compare the audited $\varepsilon$ when training gpt2 with LoRA on PersonaChat, and FFT on PersonaChat and E2E. Across all settings, the new token canary gives us better auditing performance, at the cost of slightly higher perplexity.

|  |  | new | bigram | unigram | model-based | random |
|---|---|---|---|---|---|---|
| LoRA-E2E | audit 95% | **1.24** | 0.13 | 0.37 | 0.74 | 0.13 |
|  | audit 99% | **1.01** | 0.0 | 0.20 | 0.54 | 0.0 |
|  | PPL | 4.81 | 4.73 | 4.72 | 4.74 | 4.72 |
| FFT-E2E | audit 95% | **1.04** | 0.13 | 0.37 | 0.17 | 0.13 |
|  | audit 99% | **0.86** | 0.0 | 0.20 | 0.03 | 0.0 |
|  | PPL | 4.28 | 4.23 | 4.21 | 4.23 | 4.21 |
| FFT-Pers. | audit 95% | 0.84 | **1.29** | 0.67 | 0.0 | 0.05 |
|  | audit 99% | 0.66 | **1.00** | 0.46 | 0.0 | 0.0 |
|  | PPL | 23.29 | 22.31 | 22.53 | 22.41 | 22.52 |
| LoRA-Pers. | audit 95% | **0.74** | 0.60 | 0.56 | 0.0 | 0.05 |
|  | audit 99% | **0.54** | 0.46 | 0.41 | 0.0 | 0.0 |
|  | PPL | 25.59 | 25.01 | 25.05 | 25.23 | 25.00 |
| Average | audit 95% | **0.97** | 0.54 | 0.49 | 0.23 | 0.09 |
|  | audit 99% | **0.77** | 0.37 | 0.32 | 0.14 | 0.0 |

for DP trained models at $\varepsilon = 4$. Notably, *we are able to show an empirical $\varepsilon \approx 1$ for an analytical $\varepsilon = 4$ with input space canaries and a loss-based MIA without shadow models.*

Table 4: We report the audited value of $\varepsilon$ for different models, all with the new token canary, on PersonaChat, with FFT.

|  | gpt2 | gpt2-large | gpt2-xl | Pythia-160M | Pythia-410M | qwen2.5-0.5B |
|---|---|---|---|---|---|---|
| audit 95% | 0.84 | 1.28 | 1.29 | 0.40 | 0.67 | 0.96 |
| audit 99% | 0.66 | 1.08 | 1.00 | 0.25 | 0.46 | 0.86 |
| PPL | 23.29 | 14.18 | 13.05 | 86.99 | 21.19 | 14.44 |

Table 5: The impact of training steps $T$ on privacy audit in DP trained LLMs.

|  | $T = 10$ | $T = 100$ | $T = 1000$ |
|---|---|---|---|
| audit 95% | 0 | 0.91 | 0.84 |
| audit 99% | 0 | 0.53 | 0.66 |

We present most of our results in this section on gpt2 because DP-SGD training adds memory overhead that significantly increases our training time. In Table 4 we compare auditing performance across 6 models. Interestingly, all 3 model sizes in the gpt2 family perform similarly, despite the perplexity improving significantly from gpt2 to gpt2-large.

**Our Audit Does Not Compromise Clean Accuracy.** In Table 6 we validate that our method does not significantly degrade utility on the domain specific tasks, i.e., the Personachat eval set. We compare the effect of adding our new token canaries on perplexity for both no privacy and the DPSGD training with $\varepsilon = 4$. Table 6 shows that in both cases, adding canaries to the training dataset degrades our perplexity (lower is better) by $\approx 1$. For reference, Steinke et al. (2023) report an accuracy drop of 2% due to the canaries inserted for auditing, but this is not directly comparable because they only report results on computer vision tasks. In Table 3 we observe that the new token canary degrades perplexity, while the random, unigram, and bigram canaries don't degrade perplexity. This can be

seen as a natural tradeoff between the model memorizing the canary and the model learning the clean data distribution. We don't remove the new token embedding when evaluating perplexity.

Table 6: Perplexity on PersonaChat eval set. Our method does not decrease the clean performance.

|  | no canaries | with canaries |
|---|---|---|
| no privacy | **16.1** | 16.7 |
| $\varepsilon = 4$ | **22.5** | 23.3 |

Table 7: The impact of subsampling rate $q$ on privacy audit in DP trained LLMs.

|  | $q = 0.01$ | $q = 0.1$ |
|---|---|---|
| audit 95% | 0.43 | **0.84** |
| audit 99% | 0.24 | **0.66** |

Table 8: We compare random tokens as a prefix vs test data as a prefix.

|  | Random | Test Data |
|---|---|---|
| audit 95% | **0.84** | 0.63 |
| audit 99% | **0.66** | 0.28 |

**Higher Subsampling Rate is Better for Auditing.** Prior work (Nasr et al., 2023) has shown that privacy auditing becomes substantially more difficult when the subsampling rate being audited is low. This has a significant impact on the viability of an audit, because inserting 1000 canaries into each iteration may present a nontrivial compute overhead or impact clean accuracy. Steinke et al. (2023) also use $q \geq 0.1$ for privacy auditing experiments. In Table 7 we ablate the choice of smaller subsampling rates $q$ while keeping the privacy budget constant at $\varepsilon = 4$ and training for steps $T = 1000$ for each experiment run. Similar to Nasr et al. (2023); Steinke et al. (2023), Table 7 validates the necessity of a relative large subsampling rate, i.e. $q = 0.1$ in our main results.

**Training for More Steps Improves Auditing.** Our canaries can provide a good estimation for memorization in Section 4 by approximately seeing each canary once. Our main results in DP auditing is 1000 steps with $q = 0.1$ and therefore the model approximately sees each canary 100 times. We now vary the time steps $T$ while keeping the privacy budget constant at $\varepsilon = 4$ (we add more noise at each iteration), and keeping the subsampling rate $q = 0.1$ for each experiment run. We present the results in Table 5. Table 5 shows that the one-time pass over the canary set is challenging in DP auditing and audits fails. When increasing $T$ 10 times more, i.e., $T = 100$, the DP auditing via new token canaries could achieve non-trivial results empirical $\varepsilon \approx 1$ for analytical $\varepsilon = 4$. Comparing Table 7 and Table 5, while in $(T, q) = (1000, 0.01)$ and $(T, q) = (100, 0.1)$, the models both see the canaries 10 times, the lower subsampling rate is more challenging for DP auditing.

**Random Prefixes are Better Canaries than In-Distribution Data.** We compare two approaches for selecting canary prefixes: randomly sampled tokens versus samples from the test dataset. In Table 8, we demonstrate that using random tokens as prefixes leads to more effective privacy auditing. This can be explained by considering what associations the model needs to learn during supervised fine-tuning. With test distribution prefixes, the model must balance learning two competing objectives: (1) associating the prefix with its natural, in-distribution continuations, and (2) associating it with our inserted canary token. This competition naturally reduces the probability of the model predicting the canary token. In contrast, random (OOD) prefixes only require the model to learn a single, albeit unusual, association with the canary token. This focused learning task makes the canary information more distinguishable during privacy auditing, as the model's prediction of the canary token becomes a clearer signal of memorization. This may seem like a limitation, because it means that the attacker conducting the MIA cannot get a clear signal on the in-distribution data with semantic meaning. However, in Section 4 we used samples from the test dataset as prefixes throughout and showed that when the model is not trained with DP, the attacker can correctly identify members. In the auditing threat model, we can use random prefixes for the canaries without it being a limitation for our method. However, this also shows a clear direction for future work to build on our method.

**Impact of Loss Function on Auditing Performance.** In Table 9 we find that auditing is easier when we train with an SFT objective, in line with the results in Section 4. This is because including the loss over the prefix in the MIA statistic makes the auditing test noisier, and we need very low FPR for a good audit.

Table 9: Loss over target sequence only (SFT) vs. loss over the full sequence (NWP).

|  | SFT | NWP |
|---|---|---|
| Audit 95% | **0.84** | 0.0 |
| Audit 99% | **0.66** | 0.0 |

## 6 RELATED WORK AND DISCUSSION

**Privacy Attacks in Machine Learning.** *Membership Inference* (Shokri et al., 2017; Choquette-Choo et al., 2021; Carlini et al., 2022a; Jagielski et al., 2023a), *attribute inference* (Yeom et al., 2018; Fredrikson et al., 2015), and *data extraction* (Carlini et al., 2019; 2023a;b; Biderman et al., 2023a; Tirumala et al., 2022; Mireshghallah et al., 2022; Huang et al., 2022; Lukas et al., 2023; Jagielski

et al., 2023b; Ippolito et al., 2023; Anil et al., 2023; Kudugunta et al., 2023) are the three main attacks on privacy in machine learning. Our attacks are based on membership inference, and require the logprobs of the model to compute the loss. Morris et al. (2024); Carlini et al. (2024) show that it is still possible for the attacker to access the logprobs when the logprobs are not directly available. Although we do not consider data extraction in this work, membership inference can lead to data extraction by using knowledge of the "outlier" token to iteratively guide decoding. We believe that using our method to improve existing data extraction attacks is an interesting future direction.

**Membership Inference Attacks on LLMs.** Shi et al. (2024) propose a new heuristic membership inference attack Min-K% to detect pretraining data in LLMs and provide case studied on copyright data detection, dataset contamination detection and machine unlearning verification. Kandpal et al. (2024) show that membership inference can be extended to collections of user data, their so-called "user inference", leading to stronger privacy threats on LLMs.

We are concerned with attempting to maximize the success of a membership inference attack on canary data; these works may attempt to extract data that already exists in the model. Membership inference on canaries is no less important than membership inference of real training data, because it provides us with an understanding of the *worst-case* privacy leakage. As we have discussed throughout the paper, only doing membership inference of real training data may systematically underestimate true privacy leakage, and the underlying vulnerability may only appear when training data is extracted from a production LLM (Nasr et al., 2025).

**Privacy Auditing Methods.** In this work we primarily use the method of Steinke et al. (2023) because it can do privacy auditing in one run. However, a number of privacy auditing methods have been proposed that our method is compatible with. Nasr et al. (2023) obtain tight auditing results, but require multiple runs. Pillutla et al. (2023) can re-use previous training runs to improve efficiency. Annamalai & Cristofaro (2024) exploit the model initialization for better distinguishability. Recently, Kazmi et al. (2024) propose a method for estimating privacy leakage. However, they do not provide an audit, in that they do not show a lower bound on epsilon. In the paragraph titled "Measurement Semantics" on page 6, they note: "the value PANORAMIA returns does not imply a lower bound on epsilon." In contrast, we return a provable lower bound on epsilon. To the best of our knowledge, we are the first to provide non-trivial auditing results on LLMs, as well as a systematic evaluation of the memorization rate in LLM training from the perspective of canary design.

**Privacy Preserving Language Models.** DP-SGD has been used to pretrain (Anil et al., 2021; Ponomareva et al., 2022) and fine-tune (Panda et al., 2024) LLMs. *Our work is focused on auditing any such DP training run, i.e., validate if the proposed guarantees are correct.* Orthogonal to our work are many that seek to improve DP-SGD's adoption in LLMs. These include techniques that improve compute- or memory-efficiency, such as parameter efficient techniques (Yu et al., 2022), new clipping techniques (Li et al., 2022; He et al., 2023), better hyperparameter tuning (Panda et al., 2024), and using zero-th order optimization (Tang et al., 2025). There is also DP in-context-learning (Duan et al., 2023; Wu et al., 2024; Tang et al., 2024; Hong et al., 2024) which never updates the model. Hanke et al. (2024) comprehensively evaluate the privacy-performance tradeoff of these methods.

**Discussion.** Ever since Secret Sharer (Carlini et al., 2019), work that has evaluated privacy leakage of language models via membership inference of inserted canaries has consistently found that memorization of canaries is limited. For years, this line of work showing the limited success of membership inference attacks on language models (Duan et al., 2024) has been at odds with another line of work on training data extraction from language models (Carlini et al., 2021; Nasr et al., 2025). In this work, we present a simple change in the design of the canary that vastly increases the success of MIA. This enables loss-based membership inference without shadow models, and therefore allows us to obtain the first nontrivial privacy audit of an LLM trained on real data with a realistic DP guarantee with input-space canaries. *Our work provides an efficient privacy audit that can run alongside a regular DP training run and provide a good lower bound of the privacy parameter.*

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

## A   EXPERIMENTAL DETAILS

### A.1   EXPERIMENTAL SET-UP

**Models.** We evaluate GPT2 (Radford et al., 2019) (license: mit), Pythia (Biderman et al., 2023b) (license: apache-2.0), Qwen-2.5 (Qwen-Team et al., 2024; Qwen-Team, 2024) (license: apache-2.0), Gemma (Gemma-Team et al., 2024) (license: gemma), Mistral (Jiang et al., 2023) (license: apache-2.0), and Llama3 (Team et al., 2024) (license:llama3). We outline the parameter size and tokenizer size for models we use in Tables 10 and 11.

Table 10: Model parameter and tokenizer size for GPT2 and Pythia series in our experiments.

| Model | Gpt2 | Gpt2-large | Gpt2-xl | Pythia-160m | Pythia-410m | Pythia-1.4b |
|---|---|---|---|---|---|---|
| Parameters | 124M | 774M | 1.5B | 160M | 410M | 1.4B |
| Tokenizer | | 50257 | | | 50304 | |

Table 11: Model parameter and tokenizer size for Qwen, and LLama series in our experiments.

| Model | Qwen2.5-0.5B | Qwen2.5-1.5B | Qwen2.5-3B | Llama-3.2-1B |
|---|---|---|---|---|
| Parameters | 0.5B | 1.5B | 3B | 1B |
| Tokenizer | | 151936 | | 128256 |

**Hyperparameters.** We have 1000 canaries in total. Following Steinke et al. (2023), 500 of canaries are randomly included as part of training set. We use batch size 1024 when training the models. We search lr in $[0.0001, 0.0002, 0.0005, 0.001]$ and conduct auditing on models that have the best performance, i.e., lowest perplexity. We use AdamW optimizer with default settings. For memorization evaluationg, we train for 100 steps. We use the clipping $threshold = 1$ to clip the averaged gradients in each step. For DP auditing, we train for 1000 steps. We use the clipping norm $C = 1$ for per-example clipping.

**Impact of Learning Rate on Auditing Success.** Our main results are presented with the default learning rate in Huggingface's implementation of AdamW, which is $\eta = 1e-3$. We now present results varying the learning rate. We observe that when the learning rate is larger, the model utility may drop, but we can still get good auditing performance. When we decrease the learning rate slightly, the auditing performance drops slightly. When we decrease the learning rate significantly, the utility becomes worse and the auditing performance drops to $0$. This indicates that there may be a tradeoff between DP auditing performance and performance, but we emphasize that *we are still able to obtain nontrivial auditing performance without impacting clean utility*.

Table 12: The auditing succeeds for a range of learning rates, but if the learning rate is too small then the utility and auditing performance suffer.

| Learning Rate | $1e-4$ | $5e-4$ | $1e-3$ | $5e-3$ |
|---|---|---|---|---|
| Utility | 28 | 22 | 24 | 48 |
| Audit | 0 | 0.9 | 1.3 | 1.3 |

The CDFs we visualize in Figure 3 indicate that the unigram attack will be the most effective strategy if the main criterion in attack success is how infrequent the canary token is relative to the entire training dataset. This intuition is well validated by the new token attack being the most effective by far. It also tracks the relative performance of the random, unigram, and model-based canaries as we see in Figure 1. Despite requiring knowledge of the model parameters, the model-based canary does not clearly dominate the simple unigram attack.

Table 13: Varying the LoRA rank hardly changes performance, with an AUC difference of just 0.02 between a rank of 4 and a rank of 512.

| Rank | 4 | 8 | 16 | 32 | 64 | 128 | 256 | 512 | FFT |
|------|-----|-----|-----|-----|-----|-----|-----|-----|-----|
| AUC | 0.753 | 0.763 | 0.760 | 0.773 | 0.765 | 0.774 | 0.760 | 0.774 | 0.776 |

Table 14: In the main paper we always update embeddings when we do LoRA. Without updating embeddings, neither the auditing works, nor do we get good performance.

| Type | | new | bigram | unigram | model-based | random |
|------|------|------|--------|---------|-------------|--------|
| | audit 95% | 0.74 | 0.60 | 0.56 | 0.0 | 0.05 |
| Embeddings Updated | audit 99% | 0.54 | 0.46 | 0.41 | 0.0 | 0.0 |
| | PPL | 25.59 | 25.01 | 25.05 | 25.23 | 25.00 |
| | audit 95% | 0.05 | 0 | 0 | 0 | 0.07 |
| Embeddings Frozen | audit 99% | 0 | 0 | 0 | 0 | 0 |
| | PPL | 44.88 | 29.12 | 29.28 | 29.17 | 29.30 |

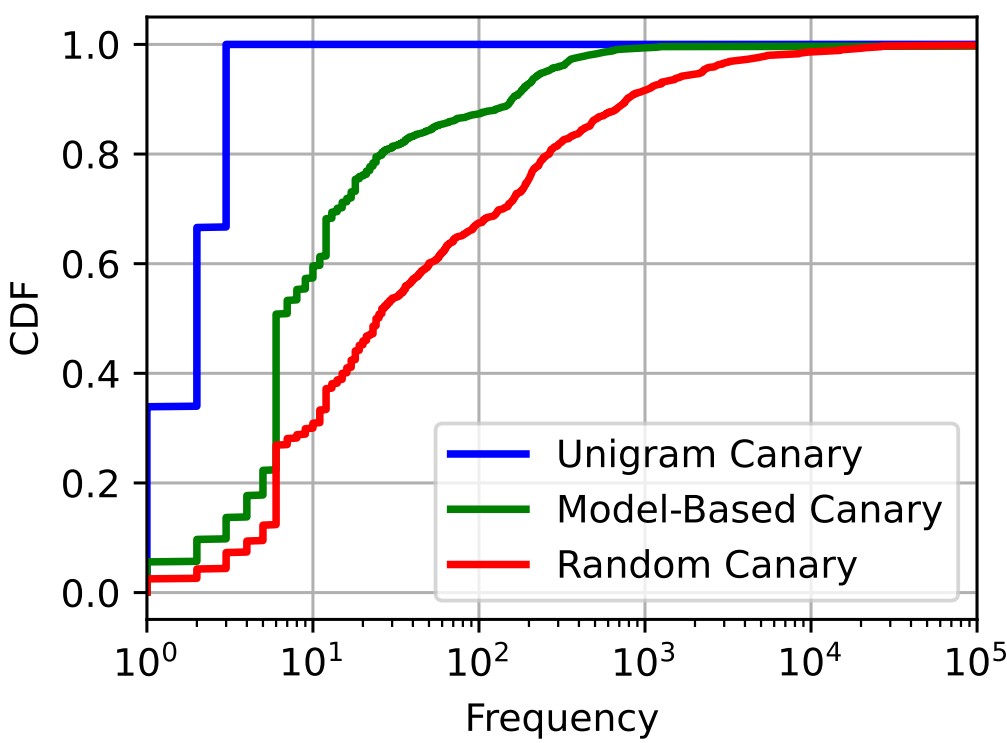

Figure 3: Frequencies of tokens selected by each strategy. By design, the unigram strategy selects the least frequent tokens.

.

