# OpenReview forum: "Privacy Auditing of Large Language Models"
_ICLR.cc/2025/Conference — ICLR 2025 Poster_

### Official Review · Reviewer_Ra2F · 2024-11-04

**Soundness:** 3
**Presentation:** 3
**Contribution:** 3
**Rating:** 8
**Confidence:** 2

**Summary:**

There is a discrepancy between the privacy leaked to attackers (usually the most extractable data) and the privacy guarantees measured by canaries (which are not as easily extractable) leading to an underestimate of true privacy leakage. Since privacy concerns itself with the worst case, LLM evaluation should be done in the worst case and so this paper present canaries that expose more privacy leakage than current methods (like random canaries). This is done by appending OOD data, for example a unigram (though generally an n-gram), to some prefix with semantic meaning. In the case of LLMs where much of the data is publicly available, it is not unreasonable for attackers to estimate distribution for the training data. In the case of LLMs it is also possible to use/craft special tokens (for which some popular LLMs already have) for the canaries. The paper shows the effectiveness of these canaries on varied LLMs in the private and non-private setting.

**Strengths:**

* The paper presents a non-trivial TPR for a low FPR, and also when compared to existing results.

* Method works in the black box setting. The paper applies this audit on LLMs

**Weaknesses:**

* The paper only evaluates one MIA and only on a single dataset.

Typos:

will easily memorized $\to$ will easily memorize

the privacy region is define as follows $\to$ ... is defined as follows.

**Questions:**

* How does the evaluation perform on other MIAs that are compatible with this privacy audit?

* How does the insertion of these different canaries compare with each other in terms of the performance of the LLMs?

* For prefix choices: why does the paper switch to randomly sampled tokens in the private setting and how is this practical if attackers are after information with semantic meaning? Similarly, how does a prefix choice of the most OOD data within the test set, or data within the test set perturbed by the DP noise perform?

---

> ### Author Response · Authors · 2024-11-23
>
> Thank you for appreciating the strong results of our paper. In this response, we want to outline how we have addressed your concerns in the revised paper.
>
> > W1: Single dataset
>
> We have added Table 1 for memorization rate estimation on E2E dataset and Table 3 for DP auditing on E2E dataset. E2E dataset is a natural language generation task that is also evaluated in prior DP NLP works Li et al. 2022 and Yu et al. 2022).
> The result of our newly added experiments on more datasets is consistent with our finding, our designed canaries, the newly added token, gives us better memorization rate estimation and dp audits than other baselines like random token.
>
> > Typos
>
> Thank you, we have fixed these.
>
> > Q1: Other MIAs
>
> As we note in Sec 2.1 and 6, Duan et al. has evaluated many different MIAs and concluded that none of them work on LLMs. There are 3 ways to make a better MIA. 1) choose data that is more separable 2) choose a better statistic 3) choose a better test based on that statistic. Prior work has already extensively explored (2) and (3) and were shown by Duan et al. to not work. Thus, we focus on (1) and show that changing this aspect alone can lead to strong privacy audits. This is the main contribution of our work, and is orthogonal to exploring (2) and (3), which may leverage our techniques in future work.
>
> > Q2: How do different canaries impact performance
>
> We have added the perplexity on the test set across canaries in Table 3. We reproduce the analysis here for convenience:
>
> In\cref{tab:main} we observe that the new token canary degrades perplexity, while the random, unigram, and bigram canaries don't degrade perplexity. This can be seen as a natural tradeoff between the model memorizing the canary and the model learning the clean data distribution.
>
> > Q3: Why do we use the random prefix for DP auditing
>
> We have added a detailed explanation of this in the paragraph accompanying Table 8 (random prefix vs test prefix). We reproduce it here for convenience:
>
> **Random Prefixes are Better Canaries than In-Distribution Data.** We compare two approaches for selecting canary prefixes: randomly sampled tokens versus samples from the test dataset. In\cref{tab:prefix}, we demonstrate that using random tokens as prefixes leads to more effective privacy auditing. This can be explained by considering what associations the model needs to learn during supervised fine-tuning. With test distribution prefixes, the model must balance learning two competing objectives: (1) associating the prefix with its natural, in-distribution continuations, and (2) associating it with our inserted canary token. This competition naturally reduces the probability of the model predicting the canary token. In contrast, random (out-of-distribution) prefixes only require the model to learn a single, albeit unusual, association with the canary token. This focused learning task makes the canary information more distinguishable during privacy auditing, as the model's prediction of the canary token becomes a clearer signal of memorization. This may seem like a limitation, because it means that the attacker conducting the MIA cannot get a clear signal on the in-distribution data with semantic meaning. However, in\cref{sec:mia_eval} we used samples from the test dataset as prefixes throughout and showed that when the model is not trained with DP, the attacker can correctly identify members. In the auditing threat model, we can use random prefixes for the canaries without it being a limitation for our method. However, this also shows a clear direction for future work to build on our method.

---

### Official Review · Reviewer_YH6v · 2024-11-04

**Soundness:** 3
**Presentation:** 3
**Contribution:** 3
**Rating:** 6
**Confidence:** 4

**Summary:**

This paper adds to the extensive body of research focused on privacy auditing in LLMs. One major issue with previous studies is that the design of the canaries leads to underestimation of the privacy leakage. The authors then developed canaries that are easy to memorize and more effective than the ones utilized in previous studies. Leveraging MIA, they performed the audits on several LLMs and showed that the attacks are more successful and hence, their method of developing canaries can be used as a standard for auditing the privacy leakage of LLMs.

**Strengths:**

While their approach for auditing LLMs is not unique, the paper provides extensive experiments to support their claims. The design of the canaries provides a better estimate of privacy risks of LLMs. Also, unlike previous studies where the insertion of canaries resulted in a decline in clean accuracy, their method maintains consistent accuracy levels.
Importantly, the paper is well written and easy to follow.

**Weaknesses:**

The obvious weakness is in the use of a single dataset, which has already been pointed out by the authors. It would be good to see how different datasets behave, especially when the canaries are inserted. Does the different dataset cause more or less leakage? This analysis would make the paper stronger

**Questions:**

I have the following concerns:
1. My major concern is that MI attack on LLMs is now a game of canary design. Previous works have shown the ineffectiveness of MI attack, as also pointed out by the authors. Hence, the authors are tying the success of MI attack to the goodness of canary design, which in some cases are not realistic / practical.
Can the authors provide guidelines, which could be considered standards to indeed follow, to effectively design the canaries?

2. Inherent to existing works on MI attack is the problem of distribution shifts [c]. How did the authors deal with the problem of distribution shift in their audit? Or does it not exisit in the current setting? While the focus of the paper is not the design of MI attack, such inherent problem could affect the privacy leakage estimation of the models.

3. The authors only considered the case of full fine-tuning, can the authors perform experiments using PEFT such as LoRA and prefix tuning?
What are the insights from using different PEFT than the current full fine-tuning?
Also, this might address the limitation of the single dataset.

4. In Figure 3, focusing on the "new" canary, what is the justification for the poor performance of pythia-1.4b, pythia-410m and gpt2-xl?

5. I would like to point the authors to a concurrent work. Although the authors used a data extraction attack and considered language models fine-tuned with PEFT, it is important to take note of this work [a] which is closely related


Related works:
[a] "Evaluating Privacy Risks of Parameter-Efficient Fine-Tuning." https://openreview.net/forum?id=i2Ul8WIQm7
[b] "Open LLMs are Necessary for Private Adaptations and Outperform their Closed Alternatives" Vincent Hanke et al. (Neurips 2024)
[c] "SoK: Membership Inference Attacks on LLMs are Rushing Nowhere (and How to Fix It)" Matthieu Meeus et al. https://arxiv.org/abs/2406.17975

---

> ### Author Response · Authors · 2024-11-23
>
> Thank you for appreciating our extensive experiments and the straightforward design of our canaries. In this response, we want to outline how we have addressed your concerns in the revised paper.
>
> > W1: Single dataset
>
> We have added results on another dataset E2E (Table 1 for memorization rate and Table 3 for DP audit). Privacy leakage across both datasets is similar. For example, at a \(99\%\) CI, the new token canary gives us an audited \(\varepsilon\) of \(0.84\) for FFT on PersonaChat, \(0.82\) for FFT on E2E, and \(0.89\) for LoRA on PersonaChat. We believe this is encouraging because our audit is of DPSGD and it should not vary much based on whether we use FFT or LoRA, or PersonaChat or E2E.
>
> > Q1: MI attack is all about canaries
>
> Canary design is in the threat model of an audit. Right now, there is no successful DP audit for LLMs. At least in our setting with the designed canaries, we know it's possible. It is an open question if we can get MIAs outside of this setting as powerful without this assumption, but the community has already agreed on the importance of audits (e.g., Steinke et al. 2024 “Privacy Auditing in O(1) Run”). We believe that if a company wants to audit the model it is training, it should use the new token canaries to get the tightest possible audit. This indicates what the privacy leakage could be in the worst case if the model were trained on a dataset where one sample contained a token not seen anywhere else, which is not entirely unrealistic.
>
> > Q2: Distribution shift
>
> In short: the problem of distribution shift does not exist in our setting and is not an issue for auditing.
>
> We have expanded the related work discussion on membership inference attacks and discussed Meeus et al. The shortcomings in membership inference attacks they identify are not an issue in our setting because we sample member and nonmember points IID from the test dataset (in section 4) or from random tokens (section 5).We now also clarify this in the setup of Section 3 as well.  Further, we only require this IID property to hold across this subset of data, not the entire pretraining dataset as in prior works.
>
> > Q3: LoRA in addition to FFT
>
> We apologize for not being clear; our results with larger models are actually all done with LoRA because we could not train, ex, Gemma-2B or anything larger with FFT. We have clarified this and explicitly added a comparison between FFT and LoRA in auditing (Table 3). As noted in the response to the first weakness, results are similar between FFT and LoRA. We also provide ablations on the LoRA rank (Table 13) and LoRA without updating embeddings (Table 14).
>
> > Q4: Poor performance on some models
>
> Results with the NWP objective have high variance because the loss is computed over the entire sequence, and the random sampling of prefixes from the test dataset means that some sequences will naturally have a higher loss before any training is done. This is the reason for the discrepancy between the NWP and SFT results in Figure 3 and Figure 4 (in the updated paper, the order is switched and we make this clear, Figure 1 is for SFT and Figure 2 is for NWP).
>
> > Q5: Concurrent work
>
> Thank you, the paper looks very relevant. Given that this is an ICLR 2025 submission, we cannot cite it, but when it is posted on Arxiv we will cite it in the camera ready.

---

> > ### Comment · Reviewer_YH6v · 2024-11-27
> > **Thanks for your response**
> >
> > Dear authors,
> >
> > Thank you for clarifying my concerns. It is well appreciated. While I had to spend extra time to see the changes in the updated manuscript, it would be better next time to color code the changes to save some time.  I have updated my score accordingly.
> >
> > **Minor:**
> >
> > For the cited papers [1-5], please use et al. after max 10 names. You don't need to list all the author names.
> >
> > [1]  Gemma: Open models based on gemini research and technology.
> >
> > [2] Palm: Scaling language modeling with pathways
> >
> > [3] The llama 3 herd of models
> >
> > [4] Llama 2: Open foundation and fine-tuned chat models.
> >
> > [5]  Qwen2 technical report
> >
> > Thanks again for the fine work

---

> > > ### Author Response · Authors · 2024-11-27
> > >
> > > Dear Reviewer YH6v
> > >
> > > Thank you for recommending to accept the paper! We apologize for not color coding the changes, and we have uploaded a version where the author lists are truncated with "et al." as you suggested. It's indeed very tiresome to scroll through 4 pages of names while trying to get to the Appendix.
> > >
> > > Thank you again for the very useful comments.

---

> ### Comment · Program_Chairs · 2024-11-24
>
> This review cites a concurrent ICLR paper [a]. The program chairs confirm that reviewer YH6v is _not_ an author of [a]. These are concurrent works and they should be evaluated independently.

---

> > ### Author Response · Authors · 2024-11-25
> >
> > We thank the PCs for their comment.
> >
> > We have updated the related work section with reference to the papers the reviewer suggested. We reproduce the relevant portion here for convenience;
> >
> > > Orthogonal to our work are many that seek to improve DP-SGD's adoption in LLMs. These include techniques that improve compute- or memory-efficiency, such as parameter efficient techniques\citep{Yu2021DifferentiallyPF}, new clipping techniques\citep{li2022large,he2023exploring}, better hyperparameter tuning\citep{panda2024new}, and using zero-th order optimization\citep{tang2024private}.
> > There is also DP in-context-learning\citep{duan2023flocks, wu2023privacypreserving, tang2024privacypreserving, hong2024dpoptmakelargelanguage} which never updates the model.
> > \citet{hanke2024open} comprehensively evaluate the privacy-performance tradeoff of these alternative methods.
> > Concurrently,\citet{anonymous2024evaluating} note that fine-tuning models with PEFT such as LoRA~(that we evaluate in\cref{tab:main}) may pose greater privacy risks than FFT, although our results do not substantiate this.

---

### Official Review · Reviewer_8DNT · 2024-11-07

**Soundness:** 2
**Presentation:** 3
**Contribution:** 3
**Rating:** 6
**Confidence:** 3

**Summary:**

This paper proposes a new approach for auditing large language models through the generation of canaries that are more informative than previous works. More precisely, several strategies for generating canaries are proposed, which are tested on several models and an auditing method is proposed that leverage these methods.

**Strengths:**

-The authors have clearly identified the exiting separation in terms of success between practical extraction attacks on large language models vs the auditing approaches that rely on the insertion of canaries that are not realistic.

-The design of the method for generating canaries is well-explained and motivated. More precisely, three different variants are proposed that all have a different rationale to generate OOD samples.

-The proposed strategies for generating canaries have been tested on a wide range of models. The experimental design is also well-motivated and described.

**Weaknesses:**

-The related work on previous approaches for the audit of LLMs is very short (only one paragraph) and thus it is not clear how the proposed approaches for generating canaries is different from these previous works. In addition, the study of Duan et al. as well as its findings should be explained in more details.

-The membership inference attack used to conduct the study (Yeom et al.) is one of the basic one and thus it is not clear if the results will directly generalize to a different membership inference attack.

-There is some redundancy in the paper that could be avoided. For instance, the objective of the auditing procedure and the fact that it aims at producing OOD sample is repeated many times.

-The auditing has been performed on GPT2 small because of computational constraints, which does not seem to be a good idea due to the low success of MIA on this model as shown in Figure 4. Rather, the proposed approach should be tested on bigger models. Overall, the auditing experiments are quite limited and should have been conducted on a wide range of models.

**Questions:**

There is currently no explanation on why the MIA does not seem to work with the GPT2 model. It would be great if the authors could provide more information about this.

Please see also the main points raised in the weakness section.

---

> ### Author Response · Authors · 2024-11-23
>
> Thank you for appreciating our well-motivated experimental design. In this response, we want to outline how we have addressed your concerns in the revised paper.
>
> > W1: Lack of related work
>
> First, we clarify that while there are many works in membership inference attacks on LLMs, we are the first privacy audit. We have expanded the related work discussion on membership inference attacks and discussed Duan et al. in detail. The shortcomings in membership inference attacks they identify are not an issue in our setting because we sample member and nonmember points IID from the test dataset (in section 4) or from random tokens (section 5).We now also clarify this in the setup of Section 3 as well.  Further, we only require this IID property to hold across this subset of data, not the entire pretraining dataset as in prior works.
>
> > W2: Only one MIA used
>
> As we note in Sec 2.1 and 6, Duan et al. has evaluated many different MIAs and concluded that none of them work on LLMs. There are 3 ways to make a better MIA. 1) choose data that is more separable 2) choose a better statistic 3) choose a better test based on that statistic. Prior work has already extensively explored (2) and (3) and were shown by Duan et al. to not work. Thus, we focus on (1) and show that changing this aspect alone can lead to strong privacy audits. This is the main contribution of our work, and is orthogonal to exploring (2) and (3), which may leverage our techniques in future work.
>
> > W3: Redundancy
>
> Thank you for pointing this out. We have reduced this.
>
> > W4: Auditing experiments need more models
>
> Thank you. Our auditing experiments now cover GPT2, GPT2-large, GPT2-xl, pythia-160m, pythia-410m, and Qwen-2-0.5B, varying model architectures and model sizes. The results are shown in Table 4 for DP auditing. This results is consistent with our finding, our designed canaries, the newly added token, gives us better DP audits than other baselines like random token.
>
> > Q1: Why does MIA not work on GPT2?
>
> Thank you for drawing our attention to this. After looking closer, we realized that this was a clerical error in the presented results, and we have updated the MIA results (see Figure 1, with corresponds to Figure 4 in the original submission). The updated MIA result for GPT2 (Figure 1) now shows that we can obtain a TPR of 0.548 at 1% FPR, which is in line with the results from other models.

---

> > ### Author Response · Authors · 2024-11-27
> >
> > Dear Reviewer 8DNT,
> >
> > Thank you again for your helpful comments, which we have incorporated into the revised manuscript. Following the recommendation of Reviewer YH6v, we have uploaded a revised version of the manuscript with the changes color-coded. We now provide specific line numbers for your ease of reading.
> >
> > > W1: Duan et al. should be discussed in more detail
> >
> > Lines 068-079 now cover this in greater detail.
> >
> > > W2: Only the MIA of Yeom et al. is used
> >
> > Lines 080-086 and Line 161 now address this point directly.
> >
> > > W3: Too much repetition of OOD
> >
> > Lines 189-193 and Lines 239-242 have been revised based on this.
> >
> > > W4: More models for auditing
> >
> > Lines 413-422 provide the results and analysis for running the audit on 6 models.
> >
> > > Q1: Why MIA doesn't work on GPT2
> >
> > Lines 324-332 explain that the NWP objective based MIA didn't work on GPT2, but does work on larger models, while the SFT objective based MIA works on all models, and Lines 381-386 explain that we are using the SFT based MIA for the DP auditing (with an ablation of this in Table 9).
> >
> > *Given that today is the last day for us to upload a revised version of the PDF, we kindly request that if there are additional changes you want to see, that you let us know today. Reviewers vRfX and YH6v requested that we add results on an additional dataset, which we have done (Lines 272-275, Lines 333-339, and Table 1 on Lines 342-347 for MIA, and then Lines 395-412 for DP auditing); consequently, they increased their scores and now recommend to accept the paper.*

---

> > > ### Author Response · Authors · 2024-12-02
> > >
> > > Dear Reviewer 8DNT,
> > >
> > > As today is the final day for reviewers to leave comments, we would like to ask whether there are any additional questions you would like us to clarify. Thank you.

---

> > > > ### Comment · Reviewer_8DNT · 2024-12-03
> > > >
> > > > Thanks for answering the issues that I have raised. I have now increased slightly my rating.

---

> > > > > ### Author Response · Authors · 2024-12-03
> > > > >
> > > > > Thank you for the response and once again for your helpful comments!

---

### Official Review · Reviewer_vRfX · 2024-11-09

**Soundness:** 3
**Presentation:** 3
**Contribution:** 3
**Rating:** 6
**Confidence:** 3

**Summary:**

The paper introduces a novel approach to privacy auditing of LLMs by designing more effective ("easy to remember") canaries compared to prior work.
These canaries, designed to be more easily memorized, improve the efficacy of existing MIAs, setting new standards in detecting privacy leakage in LLMs (for non-random train/test splits as in related work Ref[Duan et al, 2024]).
The paper claims to present the first privacy audit for LLMs in a black-box setting, demonstrating a non-trivial audit with an empirical privacy level (\epsilon~1) for models trained with a theoretical \epsilon=4.
This work highlights advancements in auditing privacy leakage without relying on training shadow models (computationally prohibitive for LLMs) or accessing intermediate iterations.

**Strengths:**

S1. Innovative contribution by proposing a new method for crafting (easy to memorize) canaries that enhance the efficacy of MIAs on LLMs and, thus, privacy audits for LLMs. This improvement in crafting canaries is generic to any MIA on LLMs, which leads to powerful MIAs.

S2. The significance/import of this topic is high and timely. By enabling more accurate and practical privacy audits, the paper advances the field's understanding of privacy leakage in LLMs and proposes a method that could become a new standard in LLM privacy assessment.

S3. The methodology is well-structured, using various canary generation methods across multiple model architectures (though only a single dataset is used).

**Weaknesses:**

*Weakness detailed below were addressed in the rebuttal*

- While justified via an "academic budget", using a single dataset limits the generability of the findings. Same comment for using a single model (GPT-2) for DP audit.

- Some works have already provided some privacy audits of LLMs. Can the authors highlight the main differences against:
[1] PANORAMIA: Privacy Auditing of Machine Learning Models without Retraining

- I also point the authors to other references on the line of research "MIAs do not work on LLMs":
[2] Inherent challenges of post-hoc membership inference for large language models
[3] Blind baselines beat membership inference attacks for foundation models
[4] Nob-MIAs: Non-biased Membership Inference Attacks Assessment on Large Language Models with Ex-Post Dataset Construction

**Questions:**

- Comparing with the privacy audit of [1], is the main contribution of this paper "first privacy audit of LLMs" still valid?

- Did the authors experiment with different training hyperparameters, canary size, or dataset configurations to assess how these might affect canary memorization?

---

> ### Author Response · Authors · 2024-11-23
>
> Thank you for appreciating our innovative contribution, potential impact, and well-structured methodology. In this response, we want to outline how we have addressed your concerns in the revised paper.
>
> > W1: Only 1 dataset and only 1 model for DP Audit
>
> We have added Table 1 for memorization rate estimation on E2E dataset and Table 3 for DP auditing on E2E dataset. E2E dataset is a natural language generation task that is also evaluated in prior DP NLP works Li et al. 2022 and Yu et al. 2022).
> We add Table 4 with 6 models total for DP auditing.
> The result of our newly added experiments on more datasets and models is consistent with our finding, our designed canaries, the newly added token, gives us better memorization rate estimation and dp audits than other baselines like random token.
>
> > W2: No Comparison to PANORAMIA
>
> Thank you for this reference. We have added this to our related work.
>
> PANORAMIA does not provide an audit, in that they do not show a lower bound on epsilon. In the paragraph titled "Measurement Semantics" on page 6, they note: "the value PANORAMIA returns does not imply a lower bound on epsilon." In contrast, we return a provable lower bound on epsilon.
>
> > W3: No Comparison to other work
>
> Thank you for these references. We have added these to our related work.
>
> [2], [3], and [4] all tackle the same question that Duan et al. (“Do Membership Inference Attacks Work on Large Language Models?”) does: if existing MIAs perform worse than “blind baselines” then they only work because of a distribution shift between members and non-members. In our work, our sampling process for member and non-member datapoints is IID across the dataset that we draw them from. We detail this dataset in each section: in Section 4, this is validation data and in Section 5, this dataset is random tokens. We now also clarify this in the setup of Section 3 as well. Therefore, the problem of distribution shifts identified in Meeus et al. and Duan et al. does not exist. This is different from prior work, which requires the IID property to hold across the entire pretraining dataset that they consider.
>
> > Q1: Is this paper still the first privacy audit of LLMs?
>
> We argue yes, because PANORAMIA [1] is not a privacy audit.
>
> > Q2: Did we vary hyperparameters
>
> Yes, we varied the learning rate (Table 12), number of training steps (Table 7), canary size (Table 2), and sampling rate (Table 6). Each of these ablations has a paragraph analyzing our results.

---

> > ### Comment · Reviewer_vRfX · 2024-11-25
> > **Thanks for the response**
> >
> > Dear authors,
> > Thank you for clarifying my questions, concerns, and differences with related work.
> > l'll increase my score accordingly.
> >
> > Best,

---

> > > ### Author Response · Authors · 2024-11-25
> > >
> > > Dear Reviewer vRfX,
> > >
> > > We appreciate your useful comments; they have greatly improved the quality of our manuscript. Thank you for raising your score, and we look forward to answering any remaining questions you may have.

---

### Author Response · Authors · 2024-11-23
**General Response**

We thank the reviewers for their time and helpful comments, which we have incorporated to improve the paper. We have uploaded a *major revision* of the paper. We added a new dataset, added more models for DP auditing, added new ablations, added more discussion of prior work, and reorganized the latter half of the paper. We would greatly appreciate the reviewers' time to assess whether our revised draft addresses the concerns.

---

### Meta-Review · Area_Chair_E2Nz · 2024-12-22

**Metareview:**

This paper presents a new approach for auditing LLMs for privacy leakage. The proposed method generates canaries in the data space by inserting new tokens, which does not rely on unrealistic assumptions used in prior work such as gradient canaries and access to model at every training iteration. Authors showed experimentally that their method can achieve much tighter lower bound of DP $\epsilon$ on a variety of models and datasets. AC believes this method can serve as an important cornerstone for future research on LLM privacy auditing and MIA, and thus recommends acceptance.

**Additional Comments On Reviewer Discussion:**

Several reviewers raised concerns regarding the use of a single model (GPT2) and dataset (PersonaChat) for evaluating privacy auditing. The authors presented additional experiment result on other models and datasets in the rebuttal, which addressed this concern.

---

### Decision · Program_Chairs · 2025-01-22

Accept (Poster)